# A Targeted Catalytic Nanobody (T-CAN) with Asparaginolytic Activity

**DOI:** 10.3390/cancers13225637

**Published:** 2021-11-11

**Authors:** Maristella Maggi, Greta Pessino, Isabella Guardamagna, Leonardo Lonati, Cristina Pulimeno, Claudia Scotti

**Affiliations:** 1Unit of Immunology and General Pathology, Department of Molecular Medicine, University of Pavia, 27100 Pavia, Italy; maristella.maggi@unipv.it (M.M.); greta.pessino01@universitadipavia.it (G.P.); 2Laboratory of Radiation Biophysics and Radiobiology, Department of Physics, University of Pavia, 27100 Pavia, Italy; isabella.guardamagna01@universitadipavia.it (I.G.); leonardo.lonati01@universitadipavia.it (L.L.); cristina.pulimeno01@universitadipavia.it (C.P.)

**Keywords:** L-asparaginase, acute lymphoblastic leukaemia, antibody-drug conjugates, CD19, catalytic nanobody

## Abstract

**Simple Summary:**

The therapy of Acute Lymphoblastic Leukemia (ALL) is based on *Escherichia coli* (*E. coli*) L-asparaginase, which is a very effective drug in most cases. However, its side effects sometimes prevent its usage or impose its interruption. The main issues derive from its bacterial origin, which elicits a strong immune response in the patient, and from its generalized action on all body compartments. In this work, we describe how we generated a fully active and miniaturized form of L-asparaginase starting from a camel single domain antibody, a class of antibodies known to have a very limited immunogenicity in humans. We also targeted it onto tumor cells by attaching it to an antibody fragment directed onto the CD19 B-cell surface receptor, expressed on ALL cells. We named this new molecule “Targeted Catalytic Nanobody” (T-CAN). The T-CAN is active and successfully binds to CD19 expressing cells in vitro. Thanks to its reduced immunogenic potential, it represents a new tool which deserves further development.

**Abstract:**

*E. coli* L-asparaginase is an amidohydrolase (EC 3.5.1.1) which has been successfully used for the treatment of Acute Lymphoblastic Leukemia for over 50 years. Despite its efficacy, its side effects, and especially its intrinsic immunogenicity, hamper its usage in a significant subset of cases, thus limiting therapeutic options. Innovative solutions to improve on these drawbacks have been attempted, but none of them have been truly successful so far. In this work, we fully replaced the enzyme scaffold, generating an active, miniaturized form of L-asparaginase by protein engineering of a camel single domain antibody, a class of antibodies known to have a limited immunogenicity in humans. We then targeted it onto tumor cells by an antibody scFv fragment directed onto the CD19 B-cell surface receptor expressed on ALL cells. We named this new type of nanobody-based antibody-drug conjugate “Targeted Catalytic Nanobody” (T-CAN). The new molecule retains the catalytic activity and the binding capability of the original modules and successfully targets CD19 expressing cells in vitro. Thanks to its theoretically reduced immunogenic potential compared to the original molecule, the T-CAN can represent a novel approach to tackle current limitations in L-asparaginase usage.

## 1. Introduction

The asparaginolytic activity is typical of a class of amidohydrolases, namely Asparaginases (EC 3.5.1.1, ASNases), found in bacteria, fungi and plants [1]. ASNases are of great clinical interest since they inhibit the growth of some tumor types by removing the asparagine and, to a lesser extent, the glutamine supply that they need to proliferate. Pharmaceutical grade ASNases are derived from *E. coli* or *Erwinia chrysanthemi* (*E. chrysanthemi*) and have been used in the clinics since the 1950s for the treatment of Acute Lymphoblastic Leukemia (ALL) [2]. ASNase-based therapy for ALL has a 90% success rate in pediatric patients, but it is still associated with several side effects and lower efficacy in adult patients [3]. Among them, one of the most limiting is its immunogenicity, more remarkable in adults than in children and dependent on the enzyme bacterial origin. A molecule recognized as “non-self” (antigen) by the immune system of the host, in fact, triggers a complex network of cells and molecules leading to the production of specific antibodies. They are able to bind the foreign molecule to facilitate its neutralization and removal, but they can also generate dangerous allergic reactions. In the case of ASNase, several epitopes have been identified as major triggers of the immune response in the patient, with the consequent need to interrupt the therapy [4,5].

Reduction of the immunogenicity of bacterial asparaginases has been attempted with different approaches. PEGylation, a strategy which consists of a chemical modification using polyethylenglycol (PEG), has been successfully used, but it has also been found to worsen some drug-related secondary side effects, such as hypertriglyceridemia and hepatotoxicity [6]. A second strategy, consisting of the loading of native asparaginase into patient-derived red blood cells (GRASPA, Erytech Pharma), recently failed clinical trials [7]. Removal of ASNase main immunogenic epitopes by rational engineering is also being attempted, but without practical impact so far [8,9,10,11].

A paradigm of successful reduction of drug immunogenicity is represented by the humanization of mouse antibodies [12] and by implementation of alternative human scaffolds engineered to become carriers of desired functionalities [13]. In this perspective, a powerful alternative to human full-length antibodies are nanobodies or single domain antibodies (sdAbs), a relatively new class of antibodies with peculiar characteristics [14]. They consist of the VH domain of antibodies which are found in camelids and puppy sharks, only consisting of two paired heavy chains. The VH molecule is very small (MW~15–20 kDa) and in some cases also able to pass the blood brain barrier (BBB) [15,16,17,18]. Nanobodies are highly compatible with the human immune system and, moreover, they can be very easily humanized [19]. The presence of three variable loops makes them attractive also from a further point of view. In fact, it has been demonstrated that catalytic properties can be either transferred from classical enzymes onto antibodies or selected ex novo. Since their discovery in 1986 [20], several abzymes (from antibody and enzyme) have been described, along with strategies for specific targeting and production [21]. To date, strategies for synthetic abzymes generation comprise site-direct mutagenesis, genetic selection, and screening by phage display libraries. It is expected that the use of abzymes built on a human-compatible scaffold could avoid immune system reactions due to the exogenous origin of protein drugs.

In this work, a camelid single domain antibody (sdAb) engineered to provide it with asparaginolytic activity is presented. The obtained sdASNase molecule has a calculated molecular weight of 14.9 KDa [22], which is nine times smaller than the original *E. coli* ASNase (MW 135 KDa), active and stable and can be successfully targeted onto cancer cells in vitro by an anti-CD19 scFv antibody fragment. We named this new antibody-drug conjugate “Targeted-Catalytic Nanobody” (T-CAN).

## 2. Materials and Methods

### 2.1. Transplant of E. coli Type II Asparaginase Catalytic Residues onto a Camelid sdAb Backbone

The amino acid sequence of a generic camel sdAb was retrieved from the Protein Data Bank (PDB), entry 1KXV (Camelid VHH Domains in Complex with Porcine Pancreatic α-Amylase). The crystallographic structure of this sdAb was then used as a template to model a novel engineered protein containing the main catalytic residues of *E. coli* type II asparaginase (EcAII). Asparaginases catalytic residues consist of two highly conserved catalytic triads (Thr-Lys-Asp and Thr-Tyr-Glu), among which the two threonines play the main role in catalysis. According to EcAII residue numbering, the residues belonging to the first and second triad are Thr12-Lys162-Asp90, and Thr89-Tyr25-Glu283, respectively. All six residues were transferred onto the sdAb backbone together with selected nearby amino acids (Figure 1). Specifically, the new sdAb sequence had the following amino acid replacements compared to the original sequence: G26T, N27K, L29T, and C30D, belonging to CDR1, and R100A, Y101E, G102V, L103N, P104Y, G105T, C106G, and P107T belonging to CDR3 (Figure 2). 

The nucleotide sequence coding for the modified sdAb (sdASNase) was electronically obtained by reverse translation and the gene was obtained by synthesis (GeneArt, Thermo Fisher Scientific, Monza, Italy). The sequence coding for the Tobacco Etch Virus (TEV) protease recognition site was inserted at the gene 3’- end, followed by 18 nucleotides encoding for 6 histidines. The obtained sequence was cloned into the pET45b(+) expression vector (Invitrogen, Thermo Fisher Scientific, Monza, Italy) between the *Nco* I and *Not* I cloning sites.

### 2.2. sdASNase Expression in E. coli BL21(DE3) ΔansA/ΔansB

*E. coli* BL21(DE3) Δ*ansA/*Δ*ansB* cells [23] transformed with the pET45b(+)-sdASNase vector by heat shock were used for recombinant protein expression. Expression of the sdASNase recombinant protein was obtained as previously described [24]. At the end of induction, cells were collected by centrifugation and the obtained pellet was stored at −80 °C. The protein was then purified from both the soluble and insoluble cytosolic fractions as described below.

### 2.3. sdASNase Purification from the Cytosolic Fraction

Cells collected after induction were let thaw on ice and resuspended in 50 mL/L culture binding buffer (50 mM Na-phosphate, 300 mM NaCl, 10 mM imidazole, pH 8.0). The cell suspension was subjected to 4 cycles of sonication/rest of 2 min each at 80% power, keeping the solution on ice all the time. After cell disruption, cellular debris were removed by centrifugation followed by filtration using a 0.22 μm filter. Cleared cell extract was applied to a 5 mL HisTrap column (Cytiva Europe, Milan, Italy) equilibrated in binding buffer. Protein elution was obtained by an imidazole step gradient from 0 mM to 500 mM. Eluted proteins were checked by SDS-PAGE electrophoresis in reducing conditions. Fractions positive for sdASNase were pooled and applied to a HiTrap Desalting column (Cytiva Europe, Milan, Italy) equilibrated in 50 mM Na-phosphate pH 7.4. The obtained fractions were loaded onto a 1 mL HiTrap Q column (Cytiva Europe, Milan, Italy) equilibrated in 50 mM Na-phosphate pH 7.4. Protein elution was obtained by a linear NaCl gradient and the fractions obtained, along with the flow-through, were analyzed by electrophoresis in reducing conditions, as previously described [24].

### 2.4. sdASNase Purification from Classical Inclusion Bodies

sdASNase purification from classical inclusion bodies was done as described by Malhawat with modifications, as previously reported [24]. After the purification of the refolded protein on a 5 mL HisTrap column (Cytiva Europe, Milan, Italy), the eluted fractions were analyzed by SDS-PAGE and fractions positive for the sdASNase protein were pooled together and concentrated on a 5 kDa cut-off concentrator. Protein concentration was determined by the μBCA assay (Pierce, Thermo Fisher Scientific, Monza, Italy).

### 2.5. Asparaginase Activity Assay

Purified sdASNase stored in Phosphate Buffered Saline (PBS) pH 7.4 was used for asparaginase activity evaluation by a spectrophotometric glutamate dehydrogenase (GADH) coupled-method described by Balcao [25]. For activity assay, the protein was concentrated at least at 2.0 mg/mL. The reaction mixture contained: 50 mM 4-(2-hydroxyethyl)piperazine-1-ethanesulfonic acid (HEPES) buffer, pH 7.5, 1 mM α-ketoglutarate (α-KG), 0.24 mM NADH, 20 U GADH and 10 mM L-asparagine (L-ASN). The reaction was started by adding at least 200 μg enzyme in a final volume of 0.5 mL. NADH consumption was recorded at 340 nm every 0.5 s using an Agilent spectrophotometer heated at 37 °C. Specific activity was determined as U/mg of protein. One Unit is defined as the amount of enzyme capable of transforming 1 μmoL of substrate into product at pH 7.5 and 37 °C in one minute.

### 2.6. Folding Stability of sdASNase at Acidic pH

The protein to be used for folding stability studies at acidic pH was loaded onto a Superdex 75 10/300 column (Cytiva Europe, Milan, Italy) equilibrated either in PBS pH 7.4 (physiological pH) or 50 mM Na_2_HPO_4_/Citric Acid pH 5.0 or 50 mM Na_2_HPO_4_/Citric Acid pH 4.0. After loading, samples were eluted by isocratic flow at 0.5 mL/min and fractions eluted at the expected volume for a globular protein of roughly 15 kDa (around 13.00 mL) were collected and pooled. The pooled fractions were incubated at 37 °C for 3.5 or 24 h. After incubation, an aliquot of each sample was re-analyzed by gel filtration to evaluate the presence of aggregates, degradation products, or denatured protein. Moreover, a sample of urea-treated sample was analyzed in denaturing condition in order to evaluate the behavior of the unfolded protein upon gel filtration. The peaks obtained for each condition (e.g., pH 5.0 and pH 4.0 at time 0, after 3.5 h and after 24 h incubation, respectively) were compared. In all cases, the gel filtration column was equilibrated in the same buffer of the analyzed sample.

### 2.7. sdASNase Cytotoxicity In Vitro

In order to evaluate sdASNase cytotoxicity in vitro, dose-response experiments were performed using an ASNase-sensitive ALL T-type leukemia cell line, MOLT-4. Cells were cultured at 37 °C in the presence of 5% CO_2_ in RPMI 1640 medium added with 10% fetal bovine serum, 2 mM L-glutamine, 100 IU/mL penicillin, and 100 μg/mL streptomycin. For tests, cells were collected by centrifugation, resuspended in fresh medium, and counted. A total of 50,000 cells/well in a volume of 100 μL were seeded and treated with 0.92, 0.46, 0.18, and 0 U/mL of sdASNase diluted in medium. The final volume in each well was 150 μL. Treated cells were incubated at 37 °C, 5% CO_2_ for 72 h. Cellular respiration was measured by 3-(4,5-dimethylthiazol-2-yl)-2,5-diphenyltetrazolium bromide (MTT) reduction assay. Cell viability was also evaluated by Trypan blue exclusion method. Cell respiration values were expressed as percentage versus the untreated control. IC_50_ value was calculated using a non-linear variable slope, four parameter interpolation method available in GraphPad Prism version 8.0.0 for Windows (GraphPad Software, San Diego, CA, USA).

### 2.8. Design and Cloning of the T-CAN Construct

The T-CAN was designed as a fusion protein comprising the sequence of the sdASNase described above, a sequence coding for a 3 × G_4_S linker, and the one coding an anti-CD19 scFv antibody fragment, derived from clone FMC63, as described in Nicholson et al. [26]. The DNA sequence of the protein was obtained by reverse translation and synthetized by GeneArt (Thermo Fisher Scientific, Monza, Italy). The gene was cloned into the pET45b(+) vector multiple cloning site (Invitrogen, Thermo Fisher Scientific, Monza, Italy) between the *Nco* I and *Not* I cloning sites. The sequence of the insert was checked by Sanger sequencing (GeneWiz, Leipzig, Germany) and the DNA construct inserted by heat-shock into *E. coli* BL21(DE3) Δ*ansA/*Δ*ansB* cells.

### 2.9. Expression and Purification of the T-CAN

Expression was obtained by inoculating a single colony into 10 mL of 2xTY medium supplemented with ampicillin (100 μg/mL) and streptomycin (50 μg/mL) at 37 °C shaking at 250 rpm overnight. The following day, 10 mL of the pre-inoculum was diluted into 500 mL medium supplied with the above antibiotics and the culture was incubated at 37 °C shaking at 250 rpm until OD600 reached 0.6 AUs. The temperature was then reduced to 25 °C and induction was performed with 1 mM IPTG for 24 h. Cells were collected by centrifugation and the soluble fraction of the pellet was extracted as described above for the sdAb. Purification was obtained by arginine extraction as reported in [24].

### 2.10. T-CAN Asparaginase Activity and Cytotoxicity Evaluation

T-CAN asparaginase activity was assayed as described above for the sdASNase. T-CAN cytotoxicity was tested on verified CD19-posivite Raji (Burkitt’s lymphoma) or REH (B-ALL) and in CD19-negative MOLT-4 cells (Appendix A) at 0.15 and 0.30 U/mL, and using commercial native ASNase (Kidrolase, Sigma-Aldrich, Merck, Milan, Italy) as a reference. MOLT-4 cells were also tested in a dose-response experiment using 0.125–1 U/mL as a concentration range to determine the IC_50_ as previously reported for sdASNase. Cell respiration was measured at 72 h by MTT-assay, as reported previously. 

### 2.11. ELISA

Purified T-CAN (11 µg in 100 µL in PBS) was immobilized on a MaxiSorp ELISA plate (Nunc) at room temperature with mild shaking for 3 h. Following incubation, the solution was removed, and each well was washed 3 times with PBS added with 0.1% *v*/*v* Tween 20 (PBS-T). Non-specific binding was prevented by blocking the well surface with 200 µL 2% *w*/*v* bovine serum albumin in PBS-T at room temperature with mild shaking for 2 h. The blocking solution was removed and recombinant h-CD19-(exons 1-4)-hFc (produced in house) was placed in contact with immobilized T-CAN for 1 h at room temperature with mild shaking. After the antigen was removed, each well was washed 3 times with PBS-T and incubated with an HRP conjugated anti-human IgG antibody (1:6000 in PBS-T, P0214 Dako A/S, Glostrup, Denmark) at room temperature for 45 min. After 3 washes, 3,3′,5,5′-tetramethylbenzidine substrate was added to each well and, after 30 min, the reaction was stopped, and reading was carried out at 450 nm in a microplate reader (Omega Polar Star, BMG Labtech, Ortenberg, Germany). Three wells containing only T-CAN and HRP conjugated anti-human IgG antibody were used as a blank.

### 2.12. T-CAN Labelling

T-CAN pure to homogeneity was used for labelling with Alexa fluor™ 488 using the Protein Labelling Kit from Invitrogen (Invitrogen, Thermo Fisher Scientific, Milan, Italy) and according to the manufacturer instructions. 

### 2.13. Cell Staining for Flow-Cytometry

Raji (Burkitt’s lymphoma), REH (B-ALL) and MOLT-4 (T-ALL) cells were collected by centrifugation, washed with ice-cold PBS, and incubated on ice with anti-hCD19 (CBL582, Merck Millipore, Milan, Italy) diluted 1:100 in PBS added with 1% *w*/*v* BSA. After three washes, cells were resuspended in 100 µL Alexa fluor 488 anti-mouse IgG (Sigma Aldrich, Merck, Milan, Italy) diluted 1:100 in PBS added with 1% *w*/*v* BSA and incubated on ice for 30 min protected from light. After washing, cells were resuspended in PBS and analyzed by flow cytometer. 

Cell death analysis was performed by seeding 250,000 cells/500 µL in 24 well plates, in the presence of 0.15 U/mL TCAN or Kidrolase or without treatment (untreated control, UT). Cells were collected after 72 h. The analysis to identify apoptotic cells and necrotic fragments was performed at 72 h after the treatment was started, following the manufacturer’s instructions for the eBioscience Annexin V Apoptosis Detection Kit-eFluor 450 (Invitrogen, Thermo Fisher Scientific, Milan, Italy). All analyses were performed with Attune NxT software v 3.1. Data were collected, applying a single gate to isolate singlets. Data analysis was performed by calculating the percentage of annexin V negative cells and setting the UT control as 100%.

### 2.14. Cell Staining for Confocal Microscopy

Cells were collected by centrifugation, washed 2 times with ice-cold PBS and fixed with 4% *v*/*v* paraformaldehyde at room temperature for 15 min. Afterwards, the solution was diluted 10 times in PBS added with 0.1% *v*/*v* Triton X100 and incubated at room temperature for 30 min. Cells were washed 2 times in PBS added with 0.5% *w*/*v* bovine albumin (PBSA), the pellet was then resuspended in ice-cold 50% *v*/*v* methanol in PBS and the cell suspension was incubated on ice for 15 min. After 3 washes in PBSA, cells were resuspended in PBSA added with 488-labelled T-CAN and incubated at room temperature with mild shaking and in the dark over-night. The next day, cells were washed 3 times with PBSA and the nuclei stained with Hoechst 33342 (Sigma Aldrich, Merck, Milan, Italy). After two final washes, cells were resuspended in a minimal volume and mounted on coverslips together with Mowiol. Images were collected using a Leica SP8 confocal microscope.

## 3. Results

### 3.1. sdASNase Expression, Purification and Specific Activity

Recombinant sdASNase was successfully purified from the cytosolic fraction by a two-step chromatographic method consisting of immobilized metal ion affinity chromatography (IMAC) and anion exchange chromatography (AEC) at pH 7.4. After IMAC, sdASNase eluted in the presence of 100 mM imidazole along with some contaminants that were successfully removed by AEC. During AEC, sdASNase eluted in the unbound fraction. Protein purity was evaluated by SDS-PAGE using a 15% acrylamide/bisacrylamide gel and resulted in >95%. The total yield of the purified protein from the cytosolic fraction resulted in 1.0–1.5 mg/L culture.

Purification from classical inclusion bodies (cIBs) was done by urea-mediated protein extraction followed by refolding by direct dilution. The method was successful in producing properly folded and functional protein that resulted pure to homogeneity (purity > 95%) after IMAC. The purification yield from cIBs was 60 times higher than the one obtained after purification from the cytosolic fraction. The recombinant sdASNase obtained through this route was used for the subsequent experiments.

Overall protein folding was evaluated by size exclusion chromatography (SEC, [27,28,29]) and the sample was eluted as a single, well-shaped peak at the expected elution volume. Protein pure to homogeneity was used to determine the sdASNase specific asparaginolytic activity in the presence of 10 mM L-ASN. The specific activity of the sdASNase resulted to be 1.09 ± 0.29 U/mg. The turn-over number or kcat (s^−1^) was calculated considering one active site per monomer (estimated molecular weight: 14.9 kDa) and resulted in 0.27 s^−1^.

### 3.2. Folding Stability of sdASNase at Acidic pH

Protein pH stability has been evaluated by size exclusion chromatography in order to determine peak shifts corresponding to protein aggregation/unfolding (peak leftward shift) or fragmentation (peak rightward shift) (Table 1).

Peaks obtained at pH 7.4, 5.0, and 4.0 at time zero were all superposable and corresponded to a volume compatible with the size of the monomeric sdASNase (roughly 13.4 mL); no other peak was evident in the chromatogram. After 3.5 h incubation at 37 °C, both at pH 5.0 and pH 4.0, a peak corresponding to the one obtained at incubation time 0 h was evident, but for the sample incubated at pH 4.0, a rightward tail and a second peak eluting at 16.99 mL were also evident. It is worthwhile mentioning that the area underneath the second peak was significantly lower than the one of the main peak eluted at 13.32 mL. Samples incubated at 37 °C at pH 5.0 and 4.0 for 24 h showed a main peak at a very similar elution volume (13.28 mL and 13.34 mL, respectively) of the corresponding samples from time 0 h (13.25 mL and 13.32 mL, respectively) corresponding to folded, globular sdASNase. In both cases, a second peak at roughly 17 mL was evident but less represented.

### 3.3. sdASNase Cytotoxicity In Vitro

Dose-response experiments on a model ASNase sensitive T-ALL cell line (MOLT-4) were performed using different sdASNase concentrations and evaluating cytotoxicity by cell respiration. The obtained IC_50_, described as the drug concentration at which 50% of cells are dead at 72 h with respect to the untreated control, resulted in 0.13 ± 0.02 U/mL, compared to the 0.002 ± 0.0004 U/mL of ASNase (Figure 3).

### 3.4. T-CAN Production, Activity and Cytotoxicity

Figure 4A (Appendix A) shows the band of the purified sdASNase fused to the anti-CD19 scFv (T-CAN). The protein MW is comprised between 37 and 50 KDa and is therefore compatible with the theoretical one (ca. 42 kDa). The purified protein retained the catalytic activity, with a specific activity even higher than the sdAb one (4.30 ± 0.26 U/mg) and an IC_50_ of 0.12 ± 0.01 versus the 0.002 ± 0.0004 U/mL of ASNase (Figure 3). Figure 4B illustrates the binding capacity of the T-CAN to CD19 through the anti-CD19 scFv moiety in an ELISA assay. 

Binding was also demonstrated on CD19 expressing REH and RAJI cells by immunostaining using 488 labelled T-CAN (Figure 5A and Appendix A).

In vitro efficacy on the same cell lines resulted in a slightly higher cytotoxicity than the one obtained by a similar number of units of the untargeted reference drug Kidrolase (percentage of live cells at 0.3 U/mL: 21.42 ± 3.60% versus 35.47 ± 3.40% for REH, respectively, and 76.66 ± 5.10% versus 89.02 ± 4.51% for RAJI cells, Figure 5B). CD19 negative cells (MOLT-4) appear to be less sensitive to the action of T-CAN both at 0.30 and 0.15 U/mL (Figure 5B and Appendix A, respectively) and preliminary results on CD19 positive and negative cell lines by the Annexin V assay at 0.15 U/mL seem to confirm this observation (Appendix A and Appendix A).

## 4. Discussion

L-asparaginase immunogenicity is still a main issue in its clinical use, as it often leads to treatment interruption, and it cannot easily be tackled by protein engineering of the wild-type molecule. An alternative approach to the problem is represented by the transplant of its catalytic site onto a completely different scaffold, selected for its desirable features. In our case, nanobodies were considered ideal for their reduced size, low immunogenicity, and potential to pass the BBB. Particularly, in the work here presented, a camelid single domain antibody (sdAb) was chosen as the minimal scaffold and engineered in order to provide it with asparaginolytic activity (sdASNase) by the transplantation of specific key residues selected from *E. coli* ASNase. The protein could be successfully produced in recombinant form by extraction from cIBs, though a certain amount could also be purified from the soluble fraction and showed a significant catalytic activity. When tested in vitro on MOLT-4 cells, the sdASNase was capable of inhibiting tumor cells growth up to 90% at ca. 1 U/mL (Appendix A, trypan blue), a dosage compatible with the one of Kidrolase used for patients [30]. It also showed to be endowed with protein resistance to acidic pH, an optimal characteristic for protein internalization via lysoendosomes, in the perspective of building a modular assembly to be internalized through a targeting moiety.

Regarding immunogenicity, VHHs generally share a high homology with human VH sequences [19]. In fact, alignment of sdASNase to the human sequences of the IMGT database (http://www.imgt.org/ (accessed on 11 February 2021)) shows that it is highly similar to the VH sequence of the IGHV3-20*04 VH gene, with a 67% identity and a 79% similarity (Appendix A). Moreover, VHHs can be humanized without loss of their stability, affinity, and specificity [19], and several VHHs are currently in clinical development [31].

Fusion of the sdASNase with a targeting anti-CD19 (FMC63 clone) scFv antibody fragment (T-CAN) shows the preservation of the ASNase activity as well as providing the possibility to target the catalytic nanobody onto CD19-expressing cells. Preliminary data also suggest a potentially slightly higher cytotoxicity than the one obtained by similar units of Kidrolase, though these results need to be confirmed.

The choice of the anti-CD19 FMC63 antibody is based on the extensive literature available on its specificity and its successful usage in clinical applications [26].

For all the reasons above, it is envisaged that the T-CAN could be used to reach the niche of tumors, both liquid and solid. Moreover, given that the T-CAN predicted reduced immunogenicity, it could represent a safer alternative for the treatment of patients with allergic reactions to the bacterial protein, first of all adult ALL patients. Beyond the well-known sensitivity of ALL cells to asparagine deprivation, there is also a relevant unmet need in AML treatment and evidence of the sensitivity of several solid tumor cell lines to asparagine deprivation (e.g., ovary, breast, lung), confirmed by clinical data in the case of pancreatic cancer [32].

In addition, as properly engineered VHHs are able to pass the BBB, this molecule could be evaluated to be used to tackle the tumoral disease located beyond the BBB, which, for example in the case of ALL, can normally be treated only with widespread radiotherapy, leading to severe side effects.

## 5. Conclusions

Our work proves that a camelid single domain antibody (sdAb) can be engineered in order to provide it with asparaginolytic activity (sdASNase). The molecule obtained has a very low molecular weight and represents a re-sized modular version of the highly successful anti-cancer protein drug ASNase. Moreover, the obtained sdASNase can be targeted onto a receptor specific for tumoral cells by an antibody fragment (T-CAN), while retaining its catalytic activity and in vitro cytotoxicity.

## 6. Patents

The T-CAN molecule is covered by PCT/IB2018/057037.

## Figures and Tables

**Figure 1 cancers-13-05637-f001:**
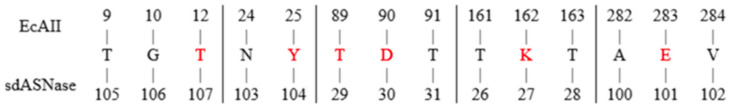
Amino acids transplanted from EcAII onto sdASNase. Top line: EcAII residue numbering; bottom line: sdASNase residue numbering. In red, residues of EcAII catalytic triads.

**Figure 2 cancers-13-05637-f002:**
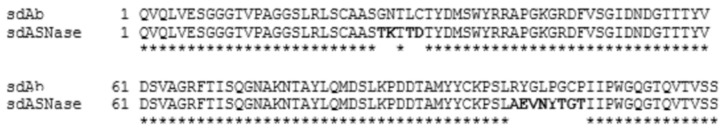
Amino acid sequence alignment of the original sdAb (from PDB ID: 1KXV, (**top**)) and newly designed sdASNase (**bottom**); in bold: transplanted amino acids; *: conserved amino acid.

**Figure 3 cancers-13-05637-f003:**
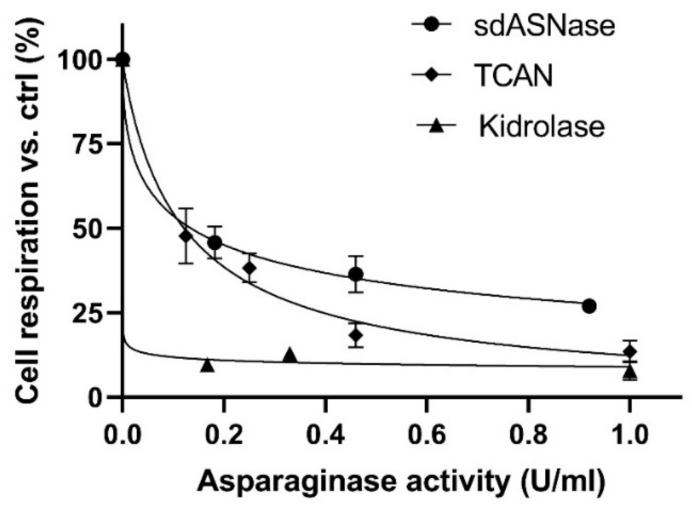
sdASNase, T-CAN and Kidrolase cytotoxicity on MOLT4 cells versus untreated control, expressed as a function of ASNase activity (U/mL) and evaluated by a cell respiration assay (MTT).

**Figure 4 cancers-13-05637-f004:**
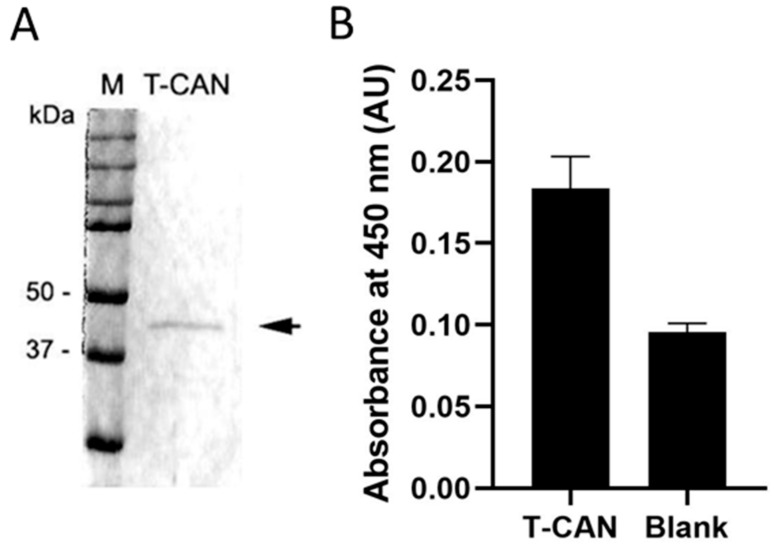
(**A**) The band of the T-CAN (expected MW: 42 kDa) is indicated by the arrow; (**B**) ELISA results of T-CAN binding to CD19.

**Figure 5 cancers-13-05637-f005:**
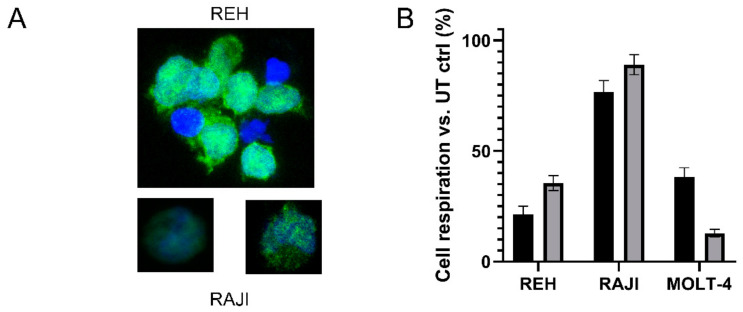
(**A**) Immunostaining of REH and RAJI cells with 488-labelled T-CAN (green). Nuclei counterstained with Hoechst (blue). (**B**) T-CAN (0.3 U/mL, black bars) cytotoxicity (MTT test) compared to that of the full-length, untargeted enzyme (Kidrolase, grey bars) on cell lines positive (REH and RAJI) or negative (MOLT-4) for CD19.

**Table 1 cancers-13-05637-t001:** Folding stability at acidic pH evaluated by SEC (Size Exclusion Chromatography).

pH	0 h		3.5 h at 37 °C		24 h at 37 °C	
	Main Peak	Further Peaks	Main Peak	Further Peaks	Main Peak	Further Peaks
7.4	13.44 mL	None	-	-	-	-
5.0	13.25 mL	None	13.32 mL	None	13.28 mL	16.72 mL
4.0	13.32 mL	18.6 mL	13.32 mL	16.99 mL	13.34 mL	17.00 mL

## Data Availability

All the data discussed in this manuscript are reported. Raw data are available from the authors upon request.

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
