# Peer review of "A Targeted Catalytic Nanobody (T-CAN) with Asparaginolytic Activity"

_cancers, 2021, doi:10.3390/cancers13225637_

Round 1
Reviewer 1 Report
The authors have addressed all my concerns and therefore I support publication without further changes.
Author Response
Reviewer 1
The authors have addressed all my concerns and therefore I support publication without further changes.
Reply
We thank the Reviewer for his/her positive comment.
Reviewer 2 Report
OVERVIEW
The authors aimed to address several issues posed by the reviewers and made improvements to the manuscript. Although the protein engineering strategy used by the authors to develop a new therapeutic tool is very interesting, unfortunately many of the experiments performed are not sufficiently controlled or not supported by statistical evidence to provide strong confidence on the therapeutic value of T-CAN. In fact, no statistical analyses were performed in the manuscript, which precludes any kind of conclusion regarding T-CAN activity on cells.
MAJOR COMMENTS
- Regarding specificity, there is no question that the CD19 antibody is very specific. However, what is not clearly shown is that addition of the anti-CD19 scFv moiety to sdASNase increases its specificity to CD19+ cells. First, the apparent increased RAJI and REH cell death comparing T-CAN to kidrolase in Figure 5C is weak and not supported by statistical tests. Moreover, in Figure 5D, kidrolase seems to be more effective in killing RAJI cells than T-CAN. Second, to show that the anti-CD19 ScFv provides targeting capacity to sdASNase, the cytotoxic action of T-CAN should be compared to sdASNase rather than kidrolase.
- I agree with Reviewer 1 that it would be very interesting to assess T-CAN effect on healthy human B cells. If T-CAN is being proposed as a novel ALL therapeutic tool, it is crucial to determine whether it can kill more efficiently malignant than healthy B cells, which also express CD19.
- S4 shows SDS-PAGE of T-CAN and not sdASNase, unlike what is stated in lines 265-269. Figure 4 also shows the same SDS-PAGE of T-CAN, so Figure S4 should be removed.
- Both Fig 3 and Fig S3 show sdASNase MOLT4 cytotoxicity, but with different results. Indeed, viability inhibition at 0.9 U/ml was stronger in S3 than 3. This needs clarification.
MINOR COMMENTS
5. Was the calculated sdASNase MW 14.9 kDa (line 87) or74 kDa (line 283)?
6. Figure 5C is not cited in the main text. Besides, Figures 5B and C appear to be 2 identical experiments, with somewhat variable results. If so, it would be better to combine the data in a single graph with 2 replicates.
7. The Reviewer 1 question about what are the black histograms in Figure S1 remains unanswered.
8. At what time were Annexin V assays performed should be stated. In addition, representative FACS plots and how the values were collected should be shown.
9. Order of citation of supplementary figures is not logical: S4 (line 269), S1 (line 332), S3 (line 352) and S2 (line 359).
Author Response
Reviewer 2
OVERVIEW
The authors aimed to address several issues posed by the reviewers and made improvements to the manuscript. Although the protein engineering strategy used by the authors to develop a new therapeutic tool is very interesting, unfortunately many of the experiments performed are not sufficiently controlled or not supported by statistical evidence to provide strong confidence on the therapeutic value of T-CAN. In fact, no statistical analyses were performed in the manuscript, which precludes any kind of conclusion regarding T-CAN activity on cells.
Reply
We thank the Reviewer for appreciating the engineering strategy and the extra work we performed, and also for the further comments. We agree that the evidence we provide in not conclusive of the therapeutic efficacy of T-CAN, as this, and possibly its advantage over Kidrolase, will be better studied with mouse experiments (see Conclusions section of the manuscript).
Regarding the new data we produced, the Reviewer is asking (see Criticism n. 1, below) for an extra control (sdASNase) that was unfortunately not specifically required during the first series of revisions and which was, therefore, not included in the new round of experiments. Moreover, Kidrolase is the reference drug used in the clinics.
Finally, though we agree that further investigation is needed to better establish TCAN therapeutic potential, we must disagree with the Reviewer on his last statement, as we indeed confirmed that sdASNase maintained ASNase activity and efficacy after fusion to an anti-CD19 targeting antibody (see Fig. 3 and line 311 of the original manuscript or 313 of the revised manuscript). For further comments, please, see also below, Criticism n. 1.
MAJOR COMMENTS
Criticism n. 1
Regarding specificity, there is no question that the CD19 antibody is very specific. However, what is not clearly shown is that addition of the anti-CD19 scFv moiety to sdASNase increases its specificity to CD19+ cells. First, the apparent increased RAJI and REH cell death comparing T-CAN to kidrolase in Figure 5C is weak and not supported by statistical tests. Moreover, in Figure 5D, kidrolase seems to be more effective in killing RAJI cells than T-CAN. Second, to show that the anti-CD19 ScFv provides targeting capacity to sdASNase, the cytotoxic action of T-CAN should be compared to sdASNase rather than kidrolase.
Reply
We thank the Reviewer for the further observations and comments. We agree that T-CAN specificity, meant as its capacity to selectively target sdASNase onto CD19+ cells, is a key point to be established for therapy, which will require a dedicated ALL xenograft mouse model. This is, however, outside the scope of this work, aimed at describing for the first time a catalytically active nanobody with asparaginolytic activity and verifying its preserved cytotoxicity after fusion with the CD19 targeting antibody (see the title of the paragraph: “3.4 T-CAN production, activity and cytotoxicity”). As mentioned above (see reply to “Overview”), this is supported both by our biochemical data and by the comparison of sdASNase and T-CAN IC50 values.
Regarding the data shown in Fig. 5, they were meant as a support, following the Reviewer’s previous requirement, to compare the cytotoxic effect of the T-CAN on both CD19+ and CD19- cells. These experiments, unfortunately, did not include sdASNase, as this request was not originally put forward by the Reviewer.
As already stated in the previous point-by-point reply (Criticism n. 2 of Reviewer 1), Fig. 5B reports data at 0.3 U/ml and Fig. 5C at 0.15 U/ml (see also Criticism n. 6 below). Fig. 5D refers instead to apoptosis (Annexin V assay, required by the Reviewer), an aspect which, we agree, requires further investigation. We opted, therefore, to include Fig. 5C and 5D as supplementary material (Fig. S2A and 2B, respectively), specifying that they represent preliminary results to be further investigated.
|
|
Figure S2 (A) and (B) MTT and Annexin V assays, respectively, performed on both RAJI and MOLT-4 cells with T-CAN (black bars) and Kidrolase (grey bars) at 0.15 U/ml. Values represents percentages versus untreated (UT) control.
For consistency and clarity, we also amended the text of the Results section and of the Discussion section like the following:
Results (page 8)
Old text:
“CD19 negative cells (MOLT-4) were confirmed to be less sensitive to the action of T-CAN (Fig. 5B). These results were also confirmed on CD19 positive and negative cell lines by the Annexin V assay (Fig. 5D).”
New text:
“CD19 negative cells (MOLT-4) appear to be less sensitive to the action of T-CAN both at 0.30 and 0.15 U/ml (Fig. 5B and Fig. S2A, respectively) and preliminary results on CD19 positive and negative cell lines by the Annexin V assay at 0.15 U/ml seem to confirm this observation (Fig. S2B and Fig. S3).”
Discussion (page 9)
Old text:
“In this case, the cytotoxicity resulted to be even higher than the one obtained by similar units of Kidrolase.”
New text:
“Preliminary data also suggest a potentially slightly higher cytotoxicity than the one obtained by similar units of Kidrolase, though these results need to be confirmed.”
Criticism n. 2
I agree with Reviewer 1 that it would be very interesting to assess T-CAN effect on healthy human B cells. If T-CAN is being proposed as a novel ALL therapeutic tool, it is crucial to determine whether it can kill more efficiently malignant than healthy B cells, which also express CD19.
Reply
We thank the Reviewer for this suggestion. As we confirmed to Reviewer 1, this is going to be our next step in efficacy evaluation of T-CAN, along with blasts derived from patients.
Criticism n. 3
S4 shows SDS-PAGE of T-CAN and not sdASNase, unlike what is stated in lines 265-269. Figure 4 also shows the same SDS-PAGE of T-CAN, so Figure S4 should be removed.
Reply
We apologise for the mistake in introducing the reference to Fig. S4. The text was amended like the following and Fig. S4, referring to an analysis originally required by the Reviewer, removed upon this further request:
Old text (page 6):
“Protein purity was evaluated by SDS-PAGE using a 15% acrylamide/bisacrylamide gel followed by ImageJ analysis (Fig S4) and resulted to be >95%.”
New text:
“Protein purity was evaluated by SDS-PAGE using a 15% acrylamide/bisacrylamide gel and resulted to be >95%.”
Criticism n. 4
Both Fig 3 and Fig S3 show sdASNase MOLT4 cytotoxicity, but with different results. Indeed, viability inhibition at 0.9 U/ml was stronger in S3 than 3. This needs clarification.
Reply
We thank the Reviewer for the observation. As already specified in the respective legends and described in the Material and Methods section, the two figures report results obtained by different methods (MTT and Trypan blue, for Fig. 3 and S4, respectively), where the MTT assay measures cell respiration and Trypan blue cell viability, respectively. The method was also mentioned in the Discussion:
Old text (page 9):
“When tested in vitro on MOLT-4 cells, the sdASNase was capable of inhibiting tumor cells growth up to 90% at ca. 1 U/ml (Fig. S2), a dosage compatible with the one of Kidrolase used for patients [30].”
New text:
“When tested in vitro on MOLT-4 cells, the sdASNase was capable of inhibiting tumor cells growth up to 90% at ca. 1 U/ml (Figure S4, Trypan blue), a dosage compatible with the one of Kidrolase used for patients [30].”
MINOR COMMENTS
Criticism n. 5
Was the calculated sdASNase MW 14.9 kDa (line 87) or 14.74 kDa (line 283)?
Reply
We confirm that the MW is 14854.35 Da. The text was amended accordingly at line 284:
Old text (page 7):
“The turn-over number or kcat (sec-1) was calculated considering one active site per monomer (estimated molecular weight: 14.74 kDa) and resulted to be 0.27 sec-1).
New text:
“The turn-over number or kcat (sec-1) was calculated considering one active site per monomer (estimated molecular weight: 14.9 kDa) and resulted to be 0.27 sec-1”.
Criticism n. 6
Figure 5C is not cited in the main text. Besides, Figures 5B and C appear to be 2 identical experiments, with somewhat variable results. If so, it would be better to combine the data in a single graph with 2 replicates.
Reply
Figure 5C was moved to the Supplementary material (see Criticism n. 1 above) and cited as Figure S2A at line 336 of the revised manuscript:
Old text (page 8):
“CD19 negative cells (MOLT-4) were confirmed to be less sensitive to the action of T-CAN (Fig. 5B). These results were also confirmed on CD19 positive and negative cell lines by the Annexin V assay (Fig. 5D).”
New text:
“CD19 negative cells (MOLT-4) appear to be less sensitive to the action of T-CAN both at 0.30 and 0.15 U/ml (Fig. 5B and Fig. S2A, respectively) and preliminary results on CD19 positive and negative cell lines by the Annexin V assay at 0.15 U/ml seem to confirm this observation (Fig. S2B and Fig. S3).”
As explained in the reply to Criticism n. 5 of the first round of revisions, Figure 5B and 5C refer to two different concentrations of ASNase (0.3 and 0.15 U/ml, respectively):
“Apoptosis was tested on RAJI and MOLT-4 cells exposed to both T-CAN and Kidrolase by the Annexin V assay, for which 0.15 U/ml were chosen to better evidence differences between the two cell lines (see Figure 3).”
To further clarify this concept, the text of the manuscript was amended as above (see Criticism n. 1) and the legend to Fig. 5 implemented:
Old text (page 8):
“Figure 5. (A) Immunostaining of REH and RAJI cells with 488-labelled T-CAN (green). Nuclei counterstained with Hoechst (blue). (B) T-CAN (black bars) cytotoxicity (MTT test) compared to that of the full-length, untargeted enzyme (Kidrolase, grey bars) on cell lines positive (REH and RAJI) or negative (MOLT-4) for CD19.”
New text:
“Figure 5. (A) Immunostaining of REH and RAJI cells with 488-labelled T-CAN (green). Nuclei counterstained with Hoechst (blue). (B) T-CAN (0.3 U/ml, black bars) cytotoxicity (MTT test) compared to that of the full-length, untargeted enzyme (Kidrolase, grey bars) on cell lines positive (REH and RAJI) or negative (MOLT-4) for CD19.”
Criticism n. 7
The Reviewer 1 question about what are the black histograms in Figure S1 remains unanswered.
Reply
The legend reporting the colour code was already present within the figure in the previously submitted version. To further clarify this point, the information was also introduced in the figure caption:
“Figure S1. CD19 immunostaining. RAJI, REH and MOLT-4 cells were analyzed by flow cytometry for the expression levels of the surface marker CD19. In black: profile of control cells, in green: profile of cells stained with an anti-CD19 antibody.”
Criticism n. 8
At what time were Annexin V assays performed should be stated. In addition, representative FACS plots and how the values were collected should be shown.
Reply
Annexin V assays were performed at 72 h after the beginning of treatment. This was introduced in the Material and methods section along with the collection method:
Old text (page 6):
“Cell death analysis was performed seeding 250.000 cells/500 µl in 24 well plates, in the presence of 0.15 U/ml TCAN or Kidrolase or without treatment (untreated control, UT). The analysis to identify apoptotic cells and necrotic fragments was performed following the manufacturer’s instructions for the eBioscience Annexin V Apoptosis Detection Kit-eFluor 450 (Invitrogen). All analyses were per-formed with Attune NxT software v 3.1.”
New text:
“Cell death analysis was performed seeding 250.000 cells/500 µl in 24 well plates, in the presence of 0.15 U/ml TCAN or Kidrolase or without treatment (untreated control, UT). Cells were collected after 72 h. The analysis to identify apoptotic cells and necrotic fragments was performed following the manufacturer’s instructions for the eBioscience Annexin V Apoptosis Detection Kit-eFluor 450 (Invitrogen). All analyses were performed with Attune NxT software v 3.1. Data were collected applying a single gate to isolate singlets. Data analysis was performed by calculating the percentage of annexin V negative cells and setting the UT control as 100%.”
Representative FACS plots were introduced as Figure S3.
Figure S3. Annexin V/7AAD analysis representative FC plots. Panels A and D: RAJI and MOLT-4 untreated control (UT), respectively. Panels B and E: RAJI and MOLT-4 treated with 0.15 U/ml TCAN, respectively. Panels C and F: RAJI and MOLT-4 treated with 0.15 U/ml Kidrolase, respectively. Cells positive for 7AAD and Annexin v (bottom right, top left and right quadrants) were considered dead. Percentage of total events for each quadrant are reported in grey.
Criticism n. 9
Order of citation of supplementary figures is not logical: S4 (line 269), S1 (line 332), S3 (line 352) and S2 (line 359).
Reply
Supplementary figures order was revised to match the text flow.

Round 2
Reviewer 2 Report
I am pleased with the final outcome and I thank the authors for their effort.